# Differential Exosomic Proteomic Patterns and Their Influence in Resveratrol Sensitivities of Glioblastoma Cells

**DOI:** 10.3390/ijms20010191

**Published:** 2019-01-07

**Authors:** Jun-Hua Nie, Hong Li, Mo-Li Wu, Xiao-Min Lin, Le Xiong, Jia Liu

**Affiliations:** 1School of Medicine, South China University of Technology (SCUT), Guangzhou 510000, China; mcnie@mail.scut.edu.cn (J.-H.N.); mcsmall@mail.scut.edu.cn (X.-M.L.); 201710106932@mail.scut.edu.cn (L.X.); 2Department of Cell Biology, College of Basic Medical Sciences, Dalian Medical University, Dalian 116044, China; lihongmcn@dlmedu.edu.cn (H.L.); moliwusx@sina.com (M.-L.W.)

**Keywords:** glioblastoma cells, resveratrol, exosome, proteomics, drug sensitivity

## Abstract

Glioblastoma multiforme (GBM) is the commonest primary brain malignancy with extremely poor prognosis. Resveratrol posseses anti-cancer effects, while GBM cells respond differently to it due to certain unknown reason(s). Because the tumor-derived exosomes are supposed to influence chemosensitivity, the exosomic proteins released from resveratrol-sensitive U251 and resveratrol-resistant glioblastoma LN428 cells are profiled before (N/Exo) and after drug treatment (Res/Exo) by label-free liquid chromatography-mass spectrometry (LC-MS). The therapeutic implications of the proteomic findings are estimated by gene ontology enrichment analysis (GO) and the Kyoto Encyclopedia of Genes and Genomes (KEGG)-based bioinformatic analyses and further elucidated by exosome co-incubating. The results reveal that U251/N/Exo but not U251/Res/Exo enhances resveratrol sensitivity of resveratrol-resistant LN428 cells. The resveratrol sensitive properties of U251 cells are not altered by either LN428/N/Exo or LN428/Res/Exo. U251/N/Exo contains higher levels of chromatin silencing and epidermis development proteins, while U251/Res/Exo has more oxygen transport and G protein-coupled receptor. Both of LN428/N/Exo and LN428/Res/Exo are rich in the proteins related with nucleosome assembly, microtubule-based process and chromatin silencing. In conclusion, U251/N/Exo sensitizes LN428 cells to resveratrol via delivering drug sensitizing signals, suggesting the presence of additional factor(s) that may determine the resveratrol sensitivities of glioblastoma cells.

## 1. Introduction

Glioblastoma multiform (GBM) is the most common primary brain malignancy with annual incidence of about 3/100,000 [1]. The standard care for GBM is maximum feasible surgical resection followed by temozolomide-based chemotherapy [2]. However, this combined therapy merely gains about 15 months of median overall survival time because of high recurrent rates and frequent radio- and chemo-resistance [3]. Moreover, the anti-glioblastoma drugs currently used cause severe toxic effects [4] and consequently reduce the quality of life of GBM patients [5]. It would therefore be of therapeutic value to explore alternative approaches for the better treatment of this lethal malignancy.

Resveratrol (trans-3,5,4′-trihydroxystibene, C_14_H_12_O_3_), the natural plant-derived compound, posses multiple anticancer activities including sensitization of radiation and chemotherapy by promoting the differentiation of cancer cells [6]. This lipophilic compound is able to cross the blood-brain barrier through simple diffusion and exerts anti-cancer effects in the brain [7]. Moreover, resveratrol exhibits little adverse effect on normal brain cells in vitro [8] and in vivo [9], indicating its safety for practical application. So far, resveratrol has not yet been clinically used because of its low in vivo bioavailability when administered systemically [10]. This therapeutic dilemma can be largely overcome via organ-targeted drug delivery [11]. For instance, lumbar puncture-delivered resveratrol efficiently suppresses rat orthotopic GBM growth [12], especially when it is applied post-operatively [13]. The above results suggest resveratrol as a potential agent against human GBMs.

It has been found that human GBM cell lines respond differently to resveratrol [14]. For instance, the human U251 cell line is highly sensitive to resveratrol in terms of distinct growth arrest and extensive apoptosis, while the LN428 cell line shows little response under the same experimental conditions [15]. These findings suggest the necessity to investigate the underlying reason(s) leading to the distinct response of GBM cells to resveratrol for personalized anti-GBM therapy. It has been supposed that the exosomes released from drug-resistance cancer cells can increase the drug-tolerance of the receptor cells including GBM cells [16]. However, no datum has been so far available concerning (1) the impact of exosomes from drug-sensitive cells in drug sensitivity and (2) the relevance of exosomic patterns with resveratrol sensitivities of GBM cells. The current study thus aims to address the above issues by profiling exosomic proteins of U251 and LN428 cells before and after resveratrol treatment and bi-directionally analyze their influence in resveratrol sensitivity.

## 2. Results

### 2.1. Distinct Response of U251 and LN428 to Resveratrol

The results of hematoxylin-eosin (H/E) morphological staining demonstrate that U251 cells showed growth arrest and extensive cell death after 100 μM resveratrol treatment for 48 h, while no distinct cell death was found in the LN428 cell population (Figure 1A). MTT cell proliferation assay (Figure 1B) revealed that OD value (0.310 ± 0.020, cell viability = 50.1%) of resveratrol-treated U251 cells was significantly reduced in comparison with that (0.618 ± 0.103, *p* < 0.01) of the untreated counterpart; the mean OD values (0.743 ± 0.047) of resveratrol-treated LN428 cells and untreated cells (0.722 ± 0.185, *p* = 0.375) have no significant different. These results indicate that U251 rather than LN428 cells were sensitive to resveratrol.

### 2.2. Prepared Exosomes from U251 and LN428 Cells without and with Drug Treatment

Hoechst DNA staining assay was used to detect mycoplasma infection and both U251 and LN428 cell lines are out of contamination. The exosomes were purified from supernatant of normally cultured U251 or LN428 cells as U251/or LN428/N/Exo, DMSO-treated as DMSO/Exo and resveratrol-treated as Res/Exo, respectively. Transmission electron microscopy (TEM) showed the presence of 30 nm to 200 nm membrane bounded vesicles (Figure 2A). In concordance, NTA revealed the exosome size distribution is from 30 nm–200 nm (Figure 2B,C). NTA-based exosome quantification showed that resveratrol promoted exosome release especially for both U215 and LN428 cells in the extents of 415.9% and 12.1%, respectively. Western blot analysis revealed that the exosome typical protein CD63 was enriched in exosome samples, while β-actin is undetectable (Figure 2D).

### 2.3. U251/N/Exo but Not U251/Res/Exo Reversed Resveratrol Resistance of LN428 Cells

Resveratrol-treated LN428 cells pre-incubated with U251/N/Exo showed significant growth suppression in comparison with their normally cultured and resveratrol-treated counterparts (Figure 3A). Exosomes from Res-treated U251 cells (U251/Res/Exo) failed to alter resveratrol resistance of LN428 (Figure 3A). The results of the MTT assay revealed a reduction of proliferation rates of U251/N/Exo- (OD = 0.624 ± 0.027) rather than U251/Res/Exo- (OD = 0.703 ± 0.047, #, *p* = 0.043) or phosphate buffered saline (PBS)-pre-incubated LN428 (OD = 0.743 ± 0.040, *, *p* = 0.011) after being treated by resveratrol (Figure 3B). The resveratrol sensitive properties of U251 (OD = 0.310 ± 0.020) remained unchanged, irrespective to LN428/N/Exo (0.0.295 ± 0.020, *p* = 0.145) or LN428/Res/Exo (0.334 ± 0.036, *p* = 0.173) pre-incubation (Figure 3C,D).

### 2.4. Distinct Protein Compositions of U251- and LN428-Derived Exosomes

The liquid chromatography-mass spectrometry (LC-MS/MS) raw data were searched against the human database. The representative proteins with significant changes (>1.3 folds) between the U251 and LN428 cells without (N/Exo) and with resveratrol treatment (Res/Exo) are shown in Table 1. A total of 123 types of protein were identified in U25 and 216 in LN428 exosomes, respectively. The detailed protein database of each group is shown in the following diagram (Figure 4A–C). U251/N/Exo have higher Keratin (KRT including KRT18, KRT19) and H2A histone family member X (H2AX including H2AFX, HIST1H2AD) levels and lower Ras-related protein 1 (Rap1 including Rap1B, Rap1A), polymeric filaments form actin (F-actin including ACTB, ACTA2) levels, and guanine nucleotide binding protein (G protein including GNAS, GNAI). In contrast, the proteomic pattern of LN428/N/Exo is distinct to that of U251/N/Exo by showing lower KRT, H2AX and higher Rap1 and F-actin. The proteomic patterns of U251 and LN428 cells are altered by resveratrol in terms of KRT, H2AX, Rap1, F-actin and G protein being conversed. 

### 2.5. Functional Classification of U251- and LN428-Derived Exosomic Proteins

By the use of gene ontology enrichment (GO) analysis, the exosome-derived proteins were characterized according to their biological process, cellular component and molecular function (Figure 5A,B). The top cellular component GO annotation was ‘extracellular exosomes’ with 103 proteins out of 131 (U251/Exo) and 178 proteins out of 216 (LN428/Exo), confirming our exosome isolation efficiency. A Kyoto Encyclopedia of Genes and Genomes (KEGG) pathway analysis of these proteins were mapped (Figure 6A,B), the Rap1 signaling and necroptosis pathway (*p* < 0.001) are significant enriched in U251/N/Exo versus U251/Res/Exo, suggesting the regulatory role of exosomes within the drug sensitivity.

### 2.6. Differential Functional Enrichment in U251/N/Exo and U251/Res/Exo

Comparative analysis reveals differential proteomic patterns between U251/N/Exo and U251/Res/Exos. In biological processes, U251/N/Exo tend to have higher expression on chromatin silencing and epidermis development, while U251/R/Exo contains more proteins involved in oxygen transport, G-protein coupled receptor signaling pathway and cellular protein metabolic process (Figure 5A). Molecular functional analysis shows that U251/N/Exo focus more on protein heterodimerization activity and DNA binding. U251/Res/Exo expression is higher than U251/N/Exo at certain precision molecular function, such as binding activity (G-protein complex binding and guanyl nucleotide binding) and GTPase activity (Figure 5A).

### 2.7. Proteomic Similarities of LN428/N/Exo and LN428/Res/Exo

Both LN428/N/Exo and LN428/Res/Exo share a number of proteins involved in the key biological processes and molecular function. The top 5 filtered biological processes were ‘nucleosome assembly’, ‘microtubule-based process’, ‘chromatin silencing’, ‘cytoskeleton organization’, and ‘oxygen transport’. Moreover, the highest rich factors in molecular functional activities are the structural constituent of the cytoskeleton and enzymatic activity (structural molecule activity, protein heterodimerization activity, GTPase activity and oxygen transport activity). The detailed protein information is shown in Figure 5B.

## 3. Discussion

Severe side effects and frequent primary and secondary drug resistance are the main challenges of anti-glioblastoma chemotherapies [27]. For instance, advanced glioblastoma patients who develop resistance to temozolomide (TMZ) have limited opportunity for further treatment [28]. Consequently, there is an urgent clinical need to explore alternative lesser toxic approaches against glioblastomas. According to our previous studies [13], resveratrol would be such a candidate because this polyphenol compound is able to cross the blood–brain barrier via simple diffusion and exerts suppressive effects on rat RG2 formed orthotopic glioblastomas without affecting the normal function of the brain [8,29]. The therapeutic outcome of resveratrol is more promising when resveratrol is lumbar punctured after partial tumor removal [13]. In the case of human glioblastoma cell lines, resveratrol efficiently inhibits the growth of U251 cells that are partially sensitive to TMZ and finally acquire TMZ resistance after a long-term treatment [30]. On the other hand, LN428 cells harboring mutant p53, p14 and p16 deletions [31] are strongly resistant to both TMZ and resveratrol [32]. These results suggest that the treatment of human glioblastomas with resveratrol should be conducted in a personalized manner. In this context, it would be of translational value to investigate the underlying mechanism(s) resulting in differential responses of glioblastoma cells to resveratrol.

Many factors are known to be related with drug resistance, of which tumor-derived exosomes (TDEs) have been given increasing attention because they participate in many physiological as well as pathological processes [33]. The relevance of TDEs to drug resistance was also proposed because of their influence in telomerase activity of neuroblastoma cells [34] and reduced cisplatin sensitivity of ovarian cancer A2780 cells [35]. Given the above evidence, it would be possible that the exosomes secreted from LN428 cells may contain drug resistant elements that presumably reverse the resveratrol-sensitive properties of U251 cells; alternatively, the exosomes delivered from U251 cells may have little influence in the resveratrol resistance of LN428 cells. These issues are addressed in the current study by pre-incubating LN428 cells with U251-derived exosomes without (U251/N/Exo) or with resveratrol treatment (U251/Res/Exo) and vice versa. The results clearly reveal that U251/N/Exo but not U251/Res/Exo apparently enhance resveratrol sensitivity of LN428 cells. These findings, for the first time, demonstrate that drug resistance of cancer cells can be reversed by the use of exosomes from the drug-sensitive cancer cells. It is also possible that certain drug-sensitizing factors are present in the exosomes of drug sensitive cells like U251/N/Exo, which may be lost upon drug treatment. In other words, the exosomic contents are altered after resveratrol treatment, irrespective of the drug sensitivities of the treated cells. The investigation of proteomic patterns of the exosomes derived from U251 and LN428 cells before and after resveratrol treatment may further ascertain this point.

It has been found that the proteins in TDEs play active roles in drug resistance by transferring HSP, P53, PTEN and APC proteins [36]. So far, the comparative data concerning the exosomic protein contents in cancer cells with different drug sensitivities remain limited. Our proteomic analyses reveal that in comparison with the protein contents of U251/Res/Exo, the higher level of H2AX that sensitizes glioblastoma cells to apoptosis [37] and the lower levels of Rap1, F-actin and G proteins related with the enhanced drug sensitivity [38,39,40] and growth suppression of cancer cells [41] are found in U251/N/Exo. Phosphorylation of H2AX is an early sign of DNA damage induced by replication stalling [42]. It has been recognized that apoptosis requires H2AX for DNA ladder formation [43], and resveratrol can promote the apoptosis pathway via phosphorylating p38 and H2AX [18]. Given the evidence of the increased H2AX and reduced Rap1 signaling is associated with proteins in the exosomes derived from resveratrol-sensitive U251 cells, it is reasonable to consider that the resveratrol resistant properties of LN428 cells can be overcome after exposure to U251/N/Exo. Conversely, the predominant presence of Rap1, F-actin and G proteins instead of H2AX in U251/Res/Exo may explain the reason of the maintained resveratrol resistance of U251/Res/Exo pre-incubated LN428 cells. It has been known that an inflammatory microenvironment is associated with the development of GBM [44]. HSP70 as an inflammatory regulating protein can activate NF-κB-iNOS-COX2-TNFα inflammatory signaling [45] and suppression of NF-κB signaling accelerates resveratrol-treated medulloblastoma cells to apoptosis without differentiation [46]. We have found that the HSP70 level is 1.5 times higher in U251/Res/Exo than that in U251/N/Exo, suggesting the increased NF-κB activating factor(s) and, therefore, lesser resveratrol sensitivity of U251/Res/Exo-incubated cells. Consequently, the enriched drug-resistant elements in U251/Res/Exo can be regarded as a protective response to rersveratrol to keep growth and survival as has been found in neuroblastoma, breast cancer and ovarian cancer cells [34,35,47].

It has been reported that drug tolerance can be induced by the exosomes of drug-resistant cell [48]. Because the drug resistant proteins are distinctly increased in U251/Res/Exo but the resveratrol-treated U251 cells are still subjected to cell crisis; the influence of exosomes secreted from resveratrol-resistant LN428 cells in resveratrol sensitivity of U251 cells is elucidated by incubating U251 cells with LN428/N/Exo and LN428/Res/Exo, respectively. The results reveal that those pre-incubated U251 cells remain sensitive to resveratrol as similar to those treated by resveratrol only (*p* > 0.05). Proteomic profiling of these two types of exosomes revealed that LN428/N/Exo and LN428/R/Exo share similar exosomic proteomic patterns in terms of the types of proteins detected and the abundance of onco-proteins such as HSP and F-actin that are in low levels in U251/N/Exo and increased in U251/Res/Exo as well [49,50]. HSP90A has been known as an exosome marker and a favorable factor for glioblastoma cell migration [51]. These results suggest that, unlike U251, resveratrol-resistant LN428 cells produce sufficient tolerance-related exosomic proteins even under normal conditions, by which they communicate drug resistant signals with each other [52]. Interestingly, neither LN428/N/Exo nor LN428/R/Exo lead U251 cells to resveratrol tolerance. Given this evidence and according to the multiple targeting feature of resveratrol, it is reasonable to consider that additional molecular events are caused by resveratrol in its sensitive cells, which would be more lethal to the treated cells such as the inactivated STAT3 signaling [53], increased oxidative stress [54] and activation of caspase-mediated apoptosis cascade [55].

In conclusion, the exosomic proteomic patterns of resveratrol-sensitive glioblastoma U251 and resveratrol-resistant LN428 cells are profiled before and after resveratrol treatment. The influences of those exosomes in resveratrol sensitivities are investigated by pre-incubating U251-derived exosomes (U251/N/Exo or U251/R/Exo) with LN428 and pre-incubating LN428-derived exosomes with (LN428/N/Exo or LN428/Res/Exo), respectively. The results clearly demonstrate that U251/N/Exo contains the higher levels of drug-sensitizing proteins and is able to reverse the resveratrol resistance of LN428 cells. U251/Res/Exo shows altered protein composition in terms of increased drug-resistant and reduced drug-sensitizing elements and, therefore, loss of the ability to reverse resveratrol resistance of LN428 cells. Although LN428/N/Exo and LN428/Res/Exo share similar exosomic proteomic patterns and are rich in onco-proteins, both of them fail to rescue U251 cells from resveratrol caused cell crisis. Our results thus suggest (1) that the exosomes from drug sensitive cells contain the proteins for a favorable therapeutic outcome; (2) that resveratrol can increase exosomic drug-resistant protein levels of the treated cells irrespective of their drug sensitivities; and (3) beyond exosomes, the presence or absence of more critical alterations determines the fate of resveratrol-treated glioblastoma cells.

## 4. Materials and Methods

### 4.1. Glioblastoma Multiform (GBM) Cell Lines and Culture

Human glioblastoma U251 cell line was obtained from the Cell Culture Facility, Chinese Academy of Sciences Cell Bank, Shanghai and LN428 cell line is generously provided by Professor Nicolas de Tribolet, Department of Neurosurgery, Central Hospital University of Laussane, Switzerland. The cells were routinely screened by Mycoplasma Stain Assay Kit (Beyotime Institute of Biotechnology, Beijing, China), and observed by fluorescence microscopy (Axio Imager 2; Carl Zeiss, Cambridge, UK). Cells were cultured in Dulbecco’s modified Eagle medium with L-glutamine (DMEM; Gibco; Thermo Fisher Scientific, Waltham, MA, USA), supplemented with 10% exosome-depleted fetal bovine serum (FBS) (ExoPerfectTM Exo-free FBS; Suer, Shanghai, China). An amount of 5 × 10^4^/mL cells were plated onto culture dishes (Corning, NY, USA) at 37 °C and 5% CO_2_ for 48 h before the experiments were performed. For haematoxylin and eosin (H/E), dozens of cell-bearing coverslips were prepared under the same experimental conditions using coverslip-preparation dishes (Jet Biofile Tech. Inc., Guangzhou, China)

### 4.2. Resveratrol Treatment and Cell Response

Resveratrol (Sigma Chem CO., St. Louis, MO, USA) was dissolved in dimethylsulfoxide (DMSO; Sigma Chem CO., St. Louis, MO, USA) to a stock concentration of 100 mM which was diluted with culture medium to the working concentration of 100 μM for cell treatment. The two GBM cell lines were treated by 100 μM resveratrol for 48 h. The cells cultured routinely were used as normal control, and the cells cultured in the medium containing 0.1% DMSO as background control. Each of the experiments was set in triplicate, and the experiments were repeated at least three times to establish confidential conclusions.

### 4.3. Evaluation of Cell Proliferation and Death

To elucidate the cellular response of U251 and LN428 cells to resveratrol, H/E morphological staining, and 3-[4,5-Dimethylthiazol-2-yl]-2,5-diphenyl-tetrazolium bromide (MTT) cell proliferation assay were performed on cell-bearing coverslips without or with the drug treatment by the methods described elsewhere [14]. Cells treated with 0.1% DMSO were used as background controls. The normally cultured cells were used as a negative control. Digital pathology scanner (Aperio CS2, Leica Biosystems, Nussloch, German) was used to observe and photograph the cells on coverslips.

### 4.4. Sample Preparation and Exosome Isolation

Exosomes were purified from supernatants from U251 (Ln428)/DMEM, U251 (LN428)/DMSO and U251 (LN428)/Res cultured in DMEM with 10% exosome-depleted FBS. The exosomes were named as N/Exo, DMSO/Exo and Res/Exo for concisely. Briefly, supernatants were purified by differential ultracentrifugation (300× *g* 10 min, 2000× *g* 20 min, 16,500× *g* 20 min, 100,000× *g* 2 h at 4 °C) on a Biosafe optima XPN-100 (Beckman Coulter, Brea, CA, USA). These rough exosomes were washed in PBS-0.22 μm syringe filter (Merck Millipore, Brulington, MA, USA), and re-centrifuge 100,000× *g* for another 2 h at 4 °C. The final pellets contents the purified exosomes were resuspended in 100 μL PBS and stored at −80 °C.

### 4.5. Transmission Electron Microscopy-Based Exosome Identification

5 μL exosome samples of the experimental groups were fixed with 1% glutaraldehyde in PBS, and a 5 μL drop of each sample was placed on a carbon-containing grid and incubated for 20 min at room temperature for electron microscopy; 5 μL of 3% phosphotungstic acid (PH = 7) was used to stain each sample for 5 min, followed by observation under a transmission electron microscope (H-7650; Hitatchi high-technologies, Tokyo, Japan) at a voltage of 80 kV.

### 4.6. Nanoparticle Tracking Analysis (NTA)-Based Exosome Quantification

Nanoparticle tracking analysis (NTA) was employed to check the prepared exosome samples. Briefly, exosomes re-suspended in 50 μL PBS were further diluted to 300-fold to achieve between 20 and 100 objects per frame and detected by NanoSight NS-300 (Malvern Panalytical, Malvern, UK). Each sample was measured in triplicate through the camera with an acquisition time of 30 s and the detection threshold was setting at 3. At least 200 completed were analyzed in each video. NTA 2.3 software was used to capture and analyze the result.

### 4.7. Protein Preparation and Western Blotting

Total exosomic and cellular protein was extracted using RIPA lysis buffer (Beyotime Institute of Biotechnology, Beijing, China). The protein concentrations of cell and exosome lysates were determined using BCA protein assay kit (Beyotime Institute of Biotechnology, Beijing, China). For Western blot analyses, the cellular and exosomic proteins (10 µg/well) were separated by electrophoresis in 10% sodium dodecylsulfate-polyacrylamide gel electrophoresis, transferred to polyvinylidene difluoride membrane (Amersham, Buckinghamshire, UK). The membrane was blocked with 5% skimmed milk in TBS-T (10 mM TrisCl, pH 8.0, 150 mM NaCl, 0.5% Tween 20) at 4 °C overnight, rinsed three times (10 min/time) with TBS-T, followed by 3 h incubation at room temperature with a mouse anti-human CD63 antibody (1:500; Santa Cruz Biotechnology, Dallas, TX, USA) or mouse anti-human β-actin antibody (1:10,000; Proteintech Group, Chicago, IL, USA), followed by 1 h incubation with HRP-conjugated rabbit anti-mouse IgG (Zymed Lab, San Francisco, CA, USA). The bound antibody was detected using Amersham Imager 600 series imagers (GE Healthcare, Chicago, IL, USA).

### 4.8. Exosome Pre-Incubation and Resveratrol Treatment

LN428 cells were incubated with the 20 μg/mL exosomes derived from U251 cells without or with 48 h 100 μM resveratrol treatment for 48 h by putting the LN428 coverslips to the 12-well plates, and the cells seeded in 96-well plates were treated in the same manner for MTT assay. U251 derived N/Exo, DMSO/Exo, Res/Exo (20 μg/mL) or PBS (vehicle), were respectively used to incubate with LN428 cells for 48 h, followed by 100 μM resveratrol or 0.1% DMSO treatment for another 48 h. The collected cell-bearing coverslips were subjected to HE staining; the cells in 96-well plates to MTT assay. U251 cells were treated in the same manners, using LN428 derived N/Exo, DMSO/Exo, Res/Exo (20 μg/mL) or PBS (vehicle), respectively.

### 4.9. Trypsin Digestion and Desalting

Tryptic digestion on exosome for each group was performed as described previously with modifications [56]; 50 μg exosome samples (detected by BCA) were precipitated by ice-cold acetone (acetone/sample 4:1) and place at −20 °C for overnight. Reducing was then undertaken in 500 µM of DTT for 1.5 h and protecting by incubation in 6.5 μL 50 mM IAA in dark for 40 min at room temperature. Tryptic digestion was done for 24 h at 37 °C to digest exosomal proteins of each experimental groups. Peptide desalting was performed (Pierce C18 Tips; Thermo Fisher Scientific, Waltham, MA, USA). Finally, the samples were eluted sequentially with 0.1% acetic acid in a 80% ACN and dispense into a Eppendorf tube. The combined eluate was concentrated in a Vacuum Concentrator (Labconco, Kansas City, MO, USA), then resuspended by 30 μL 0.1% formic acid solution and injected into an autosampler vial for further LC-MS test.

### 4.10. Liquid Chromatography-Mass Spectrometry (LC-MS/MS) Analysis

Digested exosome peptides were analyzed by reversed-phase LC on an Easy-nLc 1000 system directly accompanied with a Q Exactive Plus Orbitrap mass spectrometer (Thermo Fisher Scientific, USA) by a nanoelectrospray source (Thermo Fisher Scientific, USA). Using 20 cm length and 75 μm inner diameter to pack in house with ReproSil-Pur 130C18-AQ 3 μm particles (Dr. Maisch HPLC GmbH, Germany). The peptide mixtures were separated using 75 min linear gradients and a two-buffer system, including buffer A (0.1% FA) and buffer B (ACN/0.1% FA). The flow rate was set to 300 nL/min. Peptides effusion from the column were sprayed into the mass spectrometer with a spray voltage of 2.3 kV with a 300 °C capillary. The mass spectrometer was working in a data-dependent mode, requiring a survey scan at 70,000 resolution with a maximum injection time of 50 ms and an automatic gain control (AGC) target of 3 × 10^6^ ions. Furthermore, the scanning range is from 300 to 2000 *m*/*z*.

### 4.11. Database Search and Bioinformatic Analyses

Raw MS data were processed using Proteome Discoverer (Version 1.4.0.288, Thermo Fisher Scientific, Bremen, Germany) with the search engine SEQUEST HT. MS/MS spectra were searched against the UniprotKB human database (downloaded on, 2018). The peptide mass, fragment mass tolerance and maximum missed cleavage sties were respectively set as 10 ppm, 20 mDa and 2 sites. Moreover, a 1% false discovery rate was used to filter the identification peptides. As for quantitative analysis of MS results, we used a label-free analysis to calculate the quantitative changes of the identified proteins among different groups. To identify the most significantly differentially expressed protein, we analyzed the data through GO and KEGG pathway analyses. Proteins with expression fold change >1.3 and Student’s *t*-test, *p* < 0.05 were filtered as differentially expressed proteins between exosomes isolated from different groups.

### 4.12. Statistical Analysis

The results of cell counting and MTT assay were evaluated with the independent-samples *t*-test and analysis of variance (ANOVA). Data were presented as mean ± standard deviation (SD) of separate experiments (n ≥ 10). When required, *p*-values are stated in the figure legends (* *p* < 0.05; ** *p* < 0.01).

## Figures and Tables

**Figure 1 ijms-20-00191-f001:**
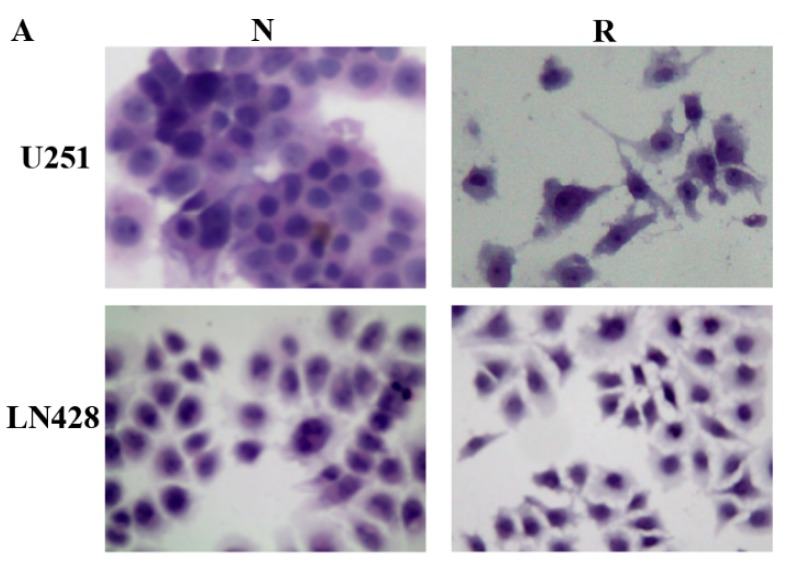
Distinct response of U251 and LN428 to resveratrol. (**A**) Hematoxylin and eosin morphological staining performed on U251 and LN428 cells without (N) or with treatment of 100 μM resveratrol (R) for 48 h (×100). Resveratrol causes growth arrest and apoptosis of U251 but not LN428 cells. (**B**) Evaluation of the cell viability of U251 and LN428 cells to resveratrol at 100 μM for 48 h by MTT assay, U251/N vs. U251/R, *, *p* = 0.4 × 10^−4^, LN428/N vs. LN428/R; #, *p* = 0.302; LN428/R vs. U251/R, $, *p* = 3 × 10^−4^.

**Figure 2 ijms-20-00191-f002:**
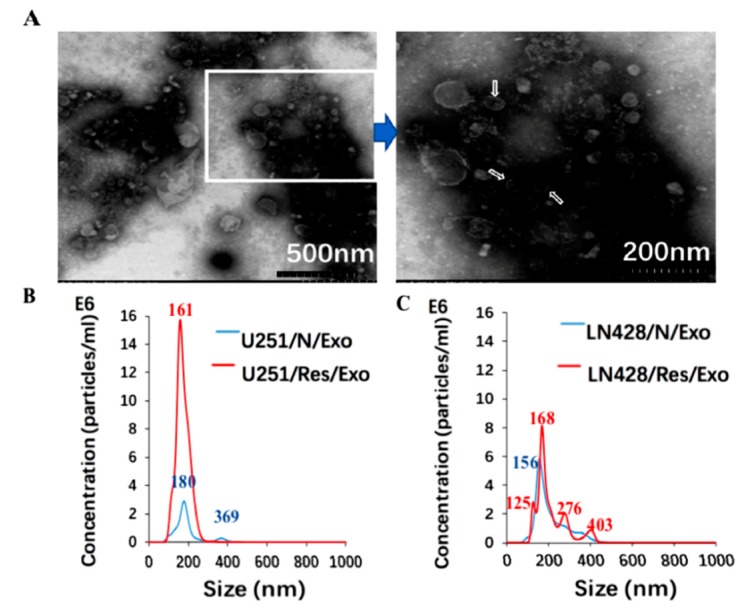
Identification of glioblastoma cell derived exosomes (Exo) purified from the supernatants by electron microscopy (**A**) and nanoparticle tracking analysis (**B**,**C**). In (**A**), the image inside the box is shown in higher magnification and the exosomes are indicated by the arrows. In (**B**,**C**), blue and red numbers indicate size of main peaks. Bar chart showing the average percentage of nanoparticles within 20–300 nm size and particle number/mL in vitro exosome preparation. Concentration and size distribution of exosomes derived from (**B**). Normal U251(U251/N) and treated U251 with resveratrol (U251/Res); (**C**). normal LN428 (LN428/N) and treating LN428 with resveratrol(LN428/Res) were measured by nanoparticle tracking analysis (NTA). Exosome concentration showed a peak at 180 nm (U251/N/Exo), 161 nm (U251/Res/Exo), 156 nm (LN428/N/Exo) and 125 nm, 168 nm (LN428/Res/Exo). (**D**). Western blot for the exosome-related proteins CD63 in U251/N/Exo, LN428/N/Exo, U251/Res/Exo and LN428/Res/Exo. The protein samples checked are positive in CD63 and negative in β-actin.

**Figure 3 ijms-20-00191-f003:**
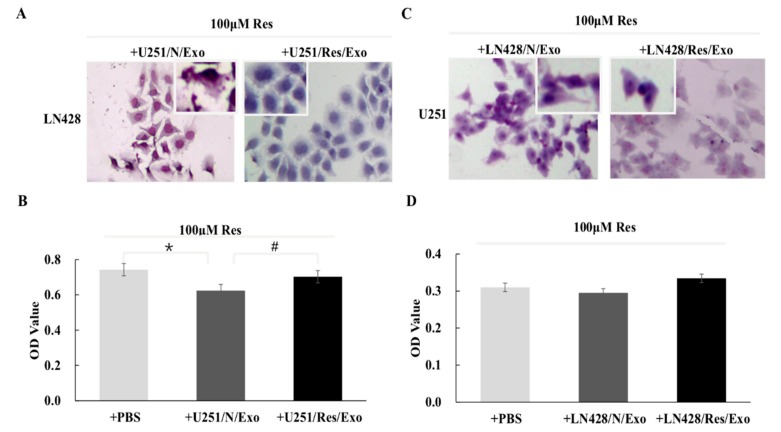
Impacts of exosomes from different origins. Hematoxylin and eosin staining and MTT assay were performed on the cell-bearing coverslips to assess resveratrol sensitivities of LN428 and U251 cells incubated with the exosomes derived from U251 and LN428 without and with resveratrol treatment. Morphology (**A**) and inhibition ratio (**B**) of LN428 cells treated by resveratrol and U251/N/Exo or U251/Res/Exo combination. The insets in (**A**) show the cells with higher magnification (×200). *, *p* = 0.011; #, *p* = 0.043. Morphology (**C**) and inhibition ratio (**D**) of U251 cells treated by resveratrol and LN428/N/Exo or LN428/Res/Exo combination. The insets in (**C**) show the cells with higher magnification (×200). Resveratrol exerts similar growth suppression effects on the three groups (*p* > 0.05).

**Figure 4 ijms-20-00191-f004:**
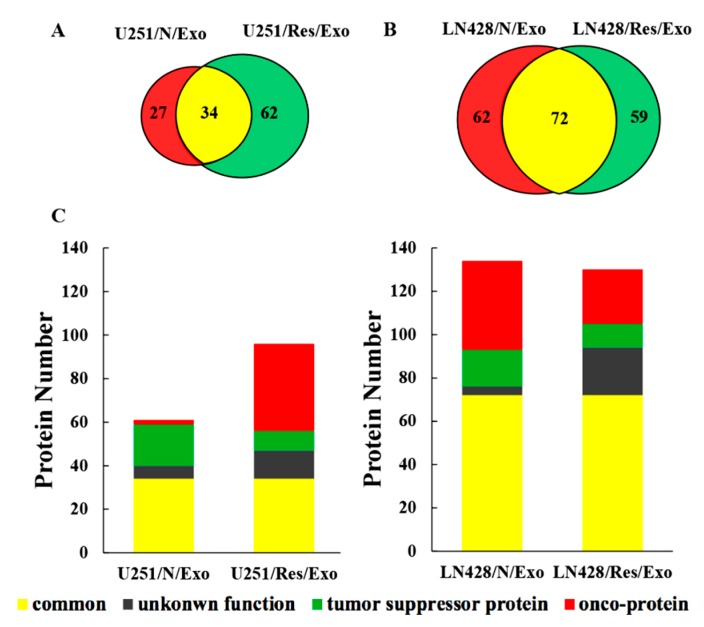
Proteomic analysis of U251 and LN428 cell-derived exosomes. (**A**,**B**) Venn diagram of liquid chromatography-mass spectrometry (LC-MS/MS)-identified proteins from U251 and LN428 cell-derived exosomes. (**C**) Subgroup numbers: Red, onco-proteins; Green, tumor suppressor proteins; Grey, proteins with unknown function; Yellow, common proteins.

**Figure 5 ijms-20-00191-f005:**
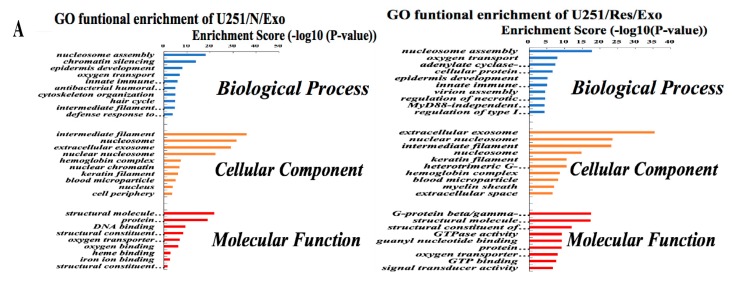
The identified proteins in U251- and LN428-derived exosomes were analyzed by gene ontology enrichment (GO) (**A**,**B**) in terms of biological process, cellular component, and molecular function annotation.

**Figure 6 ijms-20-00191-f006:**
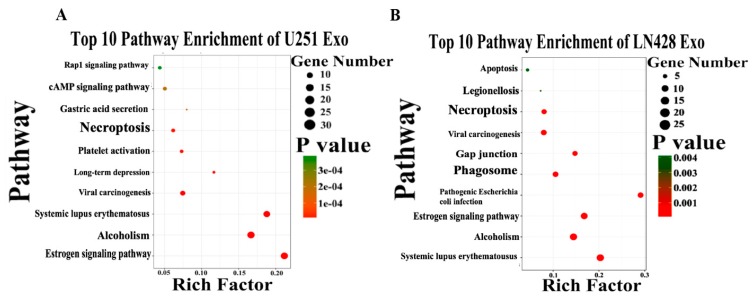
The top 10 pathway enrichment of differentially expressed exosome protein in U251 (**A**) and LN428 (**B**) cells were analyzed by Kyoto Encyclopedia of Genes and Genomes (KEGG) pathway.

**Table 1 ijms-20-00191-t001:** Representative proteins in exosomes and their quantitative changes by resveratrol in U251 and LN428 cells.

Uniprot Accession	Protein	Log_2_R *						Biological Function
		U251N/U251R	LN428N/LN428R	U251N/LN428N	U251R/LN428R	U251N/LN428R	U251R/LN428N	
**Pro-Differentiation**								
P05783	KRT18	↑↑↑ * (2.74)	↓↓ (−1.23)	↑↑ (1.55)	↓↓↓ * (−2.43)	↑ (0.31)	↓↓ (−1.20)	Promotes differentiation [17]
**Tumor-Suppressor**								
P16104	H2AX	↑↑ (1.76)	↓ (−0.44)	↑↑ (1.27)	↓↓ (−1.58)	↑ (0.18)	↓ (−0.48)	Promotes apoptosis and necroptosis [18,19] #
P02788	LTF	↑↑↑ * (2.84)	ND	ND	ND	ND	ND	Increases TMZ sensitivity [20] #
P01024	C3	ND	↑↑↑ * (2.79)	ND	ND	ND	ND	Increases TMZ and photodynamic therapy sensitivity [21] #
**Tumor-Promoting**								
P61224	RAP1B	↓ (−0.94)	↑↑ (1.75)	↓↓ (−1.17)	↑↑ (1.51)	↑ (0.58)	↓ (−0.23)	Promotes proliferation and inhibits apoptosis [22]
meP60709	ACTB	↓↓↓ * (−2.15)	↑ (0.74)	↓↓ (−1.92)	↑ (0.93)	↓↓ (−1.62)	↑ (0.63)	The same as the above [23]
Q5JWF2	GNAS	↓↓ (−1.10)	ND	ND	ND	ND	ND	The same as the above [24]
P08754	GNAI3	↓↓ (−1.10)	ND	ND	ND	ND	ND	The same as the above [24]
**Apoptosis and Necroptosis Pathway**								
Q12931	TRAP1	ND	↑↑ (1.99)	ND	ND	ND	ND	Reduces TMZ sensitivity [25] $
Q58FF8	HSP90AB2P	ND	↑↑ (1.89)	ND	ND	ND	ND	Reduces TMZ sensitivity [26] $

*, Abundance ratios of (A Exo/B Exo; ND = Not Detected; Log_2_ Ratio >2 = ↑↑↑, >1 = ↑↑, >0 = ↑, <0 = ↓, <−1 = ↓↓, <−2 = ↓↓; #, Drug sensitizing factor; $, Drug resistant factor.

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
