# Peer review of "Differential Exosomic Proteomic Patterns and Their Influence in Resveratrol Sensitivities of Glioblastoma Cells"

_ijms, 2019, doi:10.3390/ijms20010191_

Reviewer 1 Report

This manuscript entitled “Differential exosomic proteomic patterns and their influence in resveratrol sensitivities of glioblastoma cells” shows that exosomes derived from LN428 resistant to resveratrol inhibit sensitivity of U251 to resveratrol. Exosomic proteomic patterns of U251 and LN428 under resveratrol treatment are compared, and proteomic profiling and pathways are analyzed. It is suggested that resveratrol-sensitive cells derive exosome including proteins for favorable outcome and resveratrol increases exosomic proteins for drug-resistance. The results are considered of value, and may provide a clue regarding exploration of novel mechanisms of exosome in drug-sensitivity/resistance in glioblastoma. However, this reviwer considers the authors should address the following notions. 

1) The indicating results of proteomic patterns are disjointed, and key molecules or pathways involved in drug-sensitivity/resistance to resveratrol are not evident.

2) Figures 5, 6, and Discussion

Proteins and pathways identified as drug-sensitivity/resistance related factors in this study should be organized from viewpoint of the known mechanisms of reveratrol effects. It should be discussed whether these factors are specific for reveratrol treatment or they can be genelarized for the other anticancer drugs

3) Figure 2C

The photo of U251 treated with LN428/Res/Exo appears to be rather decreased in number compared with that of U251 treated with LN428/N/Exo.

4) Figure 2D

Western blot in U251 and LN428 treated with resveratrol (U251/Res and LN428/Res) also should be indicated.  

Author Response

This manuscript entitled “Differential exosomic proteomic patterns and their influence in resveratrol sensitivities of glioblastoma cells” shows that exosomes derived from LN428 resistant to resveratrol inhibit sensitivity of U251 to resveratrol. Exosomic proteomic patterns of U251 and LN428 under resveratrol treatment are compared, and proteomic profiling and pathways are analyzed. It is suggested that resveratrol-sensitive cells derive exosome including proteins for favorable outcome and resveratrol increases exosomic proteins for drug-resistance. The results are considered of value, and may provide a clue regarding exploration of novel mechanisms of exosome in drug-sensitivity/resistance in glioblastoma. However, this reviwer considers the authors should address the following notions.

1) The indicating results of proteomic patterns are disjointed, and key molecules or pathways involved in drug-sensitivity/resistance to resveratrol are not evident. 

Response:

Thanks for this comment. The biological implications of proteomic findings are further discussed as follow: Our proteomic analyses reveal that in comparison with the protein contents of U251/Res/Exo, the higher level of H2AX that sensitizes glioblastoma cells to apoptosis [37] and the lower levels of Rap1, F-actin and G proteins related with the enhanced drug sensitivity [38-40] and growth suppression of cancer cells [41] are found in U251/N/Exo.    

2) Figures 5, 6, and Discussion

Proteins and pathways identified as drug-sensitivity/resistance related factors in this study should be organized from viewpoint of the known mechanisms of reveratrol effects. It should be discussed whether these factors are specific for reveratrol treatment or they can be genelarized for the other anticancer drugs.

Response:

A very good comment. These following will be added into discussion as “Phosphorylation of H2AX is an early sign of DNA damage induced by replication stalling [42]. It has been recognized that apoptosis requires H2AX for DNA ladder formation [43], and resveratrol can promote apoptosis pathway via phosphorylating p38 and H2AX [18]. Given the evidence of the increased H2AX and reduced Rap1 signaling associated proteins in the exosomes derived from resveratrol-sensitive U251 cells, it is reasonable to consider that the resveratrol resistant properties of LN428 cells can be overcome after exposure to U251/N/Exo. Conversely, the predominant presence of Rap1, F-actin and G proteins instead of H2AX in U251/Res/Exo may explain the reason of the maintained resveratrol resistance of U251/Res/Exo pre-incubated LN428 cells.” Moreover, the contents of Table 1 have been updated in response to this comment. 

3) Figure 3C

The photo of U251 treated with LN428/Res/Exo appears to be rather decreased in number compared with that of U251 treated with LN428/N/Exo. 

Response: The cell number of U251 treated with LN428/Res/Exo and LN428/N/Exo are similar. A new image has been provided. 

4) Figure 2D

Western blot in U251 and LN428 treated with resveratrol (U251/Res and LN428/Res) also should be indicated. 

Response: Yes, the required results have been supplemented. 

Reviewer 2 Report

The current study thus aims to address the above issues by profiling exosomic proteins of U251 and LN428 cells before and after resveratrol treatment and bi-directionally analyse their influence in resveratrol sensitivity. The information and data in this manuscript are quite abundant to be considered to accept. However, please revised the following minor comments before fully accepted.

Please describe N and R within your figure legend 1.

It is better to change all * (with multiply meaning) symbol into ×.

Please confirm sentence 79-81 ‘’NTA-based exosome quantification showed that resveratrol promoted exosome releasing especial for both U215 and LN428 cells in the extents of 415.9% and 12.1% ‘’ whether 415.9% is correct the number or typo.

Please add space between number and unit in whole manuscript, for example 5 μl but not 5μl.

Author Response

The current study thus aims to address the above issues by profiling exosomic proteins of U251 and LN428 cells before and after resveratrol treatment and bi-directionally analyse their influence in resveratrol sensitivity. The information and data in this manuscript are quite abundant to be considered to accept. However, please revised the following minor comments before fully accepted.

Please describe N and R within your figure legend 1.

Response: Yes, it has been added. 

(A) Hematoxylin and eosin morphological staining performed on U251 and LN428 cells without (N) or with treatment of 100 mM resveratrol (R) for 48 h (X100). Resveratrol causes growth arrest and apoptosis of U251 but not LN428 cells.”

It is better to change all * (with multiply meaning) symbol into ×. 

Response: Yes, been corrected.

Please confirm sentence 79-81 ‘’NTA-based exosome quantification showed that resveratrol promoted exosome releasing especial for both U215 and LN428 cells in the extents of 415.9% and 12.1% ‘’ whether 415.9% is correct the number or typo.

Response: Many thanks. The typos have been corrected as “NTA-based exosome quantification showed that resveratrol promoted exosome release especial for U215 cells (415.9%) rather than LN428 cells (12.1%), respectively.”

Please add space between number and unit in whole manuscript, for example 5 μl but not 5μl.

Response: Yes, been corrected.

Reviewer 3 Report

Dear Editor,

I send you my review comments on the article “Differential Exosomic Proteomic Patterns and Their Influence in Resveratrol Sensitivities of Glioblastoma cells” (Jun-Hua Nie et al.) The Authors presented data about the exosomic proteomic patterns of resveratrol-sensitive glioblastoma U251 and resistant LN428 cells.

In my opinion, the work of Jun-Hua Nie et al. is adequately developed. The manuscript title entirely represent data enclosed, and overall the topic is of interest. Statistical analysis appears suitable for the purpose of the study.

Minor issues:

-          Please check through the manuscript the reported reference. Often the space between the last word and the bracket is missing. Please verify.

-          To better clarify the purpose of the study, please include more details on the glioma cell lines used (e.g. resistance to standard chemotherapeutic drugs).

-          Have the Authors evaluate the exosomic proteomic patterns (with or without resveratrol treatment) in glioma cell lines resistant to Temozolomide (TMZ), the common drug used for glioma treatment? For example, glioma cell lines T98G and U138 are highly resistant to TMZ.

-          It is known that glioma is characterized by an inflammatory microenvironment. Based on this fact, have the Authors studied some inflammatory markers (such as COX-2, NOS2, etc) following resveratrol exposure? Resveratrol decreases the expression of genes involved in inflammation through transcriptional regulation. It would be interesting to know if the exosomic derived-proteins involved in the inflammatory process are modulated by resveratrol.

-          An important advancement in glioma research is the identification of a subset of cells in the tumor mass with stem cell properties called glioma stem cells (GSCs). GSCs demonstrate an ability to proliferate indefinitely, differentiate into neuronal and glial lineages, that may underlie the initiation, propagation, recurrence, and therapeutic resistance of the tumor. Have the Authors analysed the exosomic cargo of these cells? In order to improve the understanding of the manuscript, in the Discussion section, Authors should be speculate about the inflammatory environment of glioma and the presence of glioma stem cells and their possible treatment with resveratrol. In an article has been simultaneously demonstrated both the NOS2 expression, an inflammatory marker, in U251 cell line kept both in normal condition (adherent cells) and their ability to generate neurospheres (glioma stem cells) (NOS2 expression in glioma cell lines and glioma primary cell cultures: correlation with neurosphere generation and SOX-2 expression. Palumbo P. Oncotarget. 2017;8(15):25582-25598. doi: 10.18632/oncotarget.16106.). Another work studied the effect of resveratrol on GSCs. It was able to inhibit cell proliferation, increase cell mortality, and strongly decrease cell motility, modulating the Wnt signaling pathway and the EMT activators (Resveratrol Impairs Glioma Stem Cells Proliferation and Motility by Modulating the Wnt Signaling Pathway. Chiara Cilibrasi C. PLoS One. 2017; 12(1): e0169854 Published online 2017 Jan 12. doi: [10.1371/journal.pone.0169854]).

I consider that the manuscript can be accepted for publication on IJMS, however, the suggestions above reported should be inserted before publication.

Author Response

Dear Editor,

I send you my review comments on the article “Differential Exosomic Proteomic Patterns and Their Influence in Resveratrol Sensitivities of Glioblastoma cells” (Jun-Hua Nie et al.) The Authors presented data about the exosomic proteomic patterns of resveratrol-sensitive glioblastoma U251 and resistant LN428 cells.

In my opinion, the work of Jun-Hua Nie et al. is adequately developed. The manuscript title entirely represent data enclosed, and overall the topic is of interest. Statistical analysis appears suitable for the purpose of the study.

Minor issues:

-  Please check through the manuscript the reported reference. Often the space between the last word and the bracket is missing. Please verify.

Response: Yes, the references have been re-checked and updated.

-   To better clarify the purpose of the study, please include more details on the glioma cell lines used (e.g. resistance to standard chemotherapeutic drugs).

Response: Yes, more details of the two cell lines used in this study have been provided as U251 cells that are partially sensitive to TMZ and finally acquire TMZ resistance after a long-term treatment [30]. On the other hand, LN428 cells harboring mutant p53, p14 and p16 deletions [31] are strongly resistance to both TMZ  and resveratrol [32].”

-   Have the Authors evaluate the exosomic proteomic patterns (with or without resveratrol treatment) in glioma cell lines resistant to Temozolomide (TMZ), the common drug used for glioma treatment? For example, glioma cell lines T98G and U138 are highly resistant to TMZ. 

Response: Good comment. According to our data of an ongoing study, LN428 cell line is highly resistant to resveratrol, TMZ as well as their combination.

-  It is known that glioma is characterized by an inflammatory microenvironment. Based on this fact, have the Authors studied some inflammatory markers (such as COX-2, NOS2, etc) following resveratrol exposure? Resveratrol decreases the expression of genes involved in inflammation through transcriptional regulation. It would be interesting to know if the exosomic derived-proteins involved in the inflammatory process are modulated by resveratrol.

Response: In this study, we have found that the inflammatory regulating protein HSP70 is in lower level in U251/N/Exo than U251/Res/Exo. HSP70 is positive related to the NF-κB-iNOS-COX2-TNF α inflammatory signaling pathway, and it will be discussed in detail in the final question. Our previous report demonstrated that resveratrol-suppressed medulloblastoma cells showed enhanced NF-κB signaling activity in response to the increased apoptosis pressure and STAT3 inactivation (Wen S et al. J Neuro-oncol, 2011, 104:169-177).

-     An important advancement in glioma research is the identification of a subset of cells in the tumor mass with stem cell properties called glioma stem cells (GSCs). GSCs demonstrate an ability to proliferate indefinitely, differentiate into neuronal and glial lineages, that may underlie the initiation, propagation, recurrence, and therapeutic resistance of the tumor. Have the Authors analysed the exosomic cargo of these cells? In order to improve the understanding of the manuscript, in the Discussion section, Authors should be speculate about the inflammatory environment of glioma and the presence of glioma stem cells and their possible treatment with resveratrol. In an article has been simultaneously demonstrated both the NOS2 expression, an inflammatory marker, in U251 cell line kept both in normal condition (adherent cells) and their ability to generate neurospheres (glioma stem cells) (NOS2 expression in glioma cell lines and glioma primary cell cultures: correlation with neurosphere generation and SOX-2 expression. Palumbo P. Oncotarget. 2017;8(15):25582-25598. doi: 10.18632/oncotarget.16106.). Another work studied the effect of resveratrol on GSCs. It was able to inhibit cell proliferation, increase cell mortality, and strongly decrease cell motility, modulating the Wnt signaling pathway and the EMT activators (Resveratrol Impairs Glioma Stem Cells Proliferation and Motility by Modulating the Wnt Signaling Pathway. Chiara Cilibrasi C. PLoS One. 2017; 12(1): e0169854 Published online 2017 Jan 12. doi: [10.1371/journal.pone.0169854]). 

Response: A very good suggestion. These following has been added in the discussion It has been known that an inflammatory microenvironment is associated with the development of GBM [44]. HSP70 as an inflammatory regulating protein can activate NF-κB-iNOS-COX2-TNFα inflammatory signaling [45] and suppression of NF-κB signaling accelerates resveratrol-treated medulloblastoma cells to apoptosis without differentiation [46]. We have found that HSP70 level is 1.5 times higher in U251/Res/Exo than that in U251/N/Exo, suggesting the increased NF-κB activating factor(s) and, therefore, lesser resveratrol sensitivity of U251/Res/Exo-incubated cells.

Round  2

Reviewer 1 Report

This manuscript has been improved. This reviwer would  expect the authors to clarify the molecular mechanism underlying exosomic proteomic change in reveratrol sensitiviy/resistance in future study.